# ERGO: Breaking Down the Wall between Human Health and Environmental Testing of Endocrine Disrupters

**DOI:** 10.3390/ijms21082954

**Published:** 2020-04-22

**Authors:** Henrik Holbech, Peter Matthiessen, Martin Hansen, Gerrit Schüürmann, Dries Knapen, Marieke Reuver, Frédéric Flamant, Laurent Sachs, Werner Kloas, Klara Hilscherova, Marc Leonard, Jürgen Arning, Volker Strauss, Taisen Iguchi, Lisa Baumann

**Affiliations:** 1Department of Biology, University of Southern Denmark, 5230 Odense M, Denmark; 2Matthiessen Consultancy, Dolfan Barn, Beulah LD5 4UE, UK; petermatthiessen12@gmail.com; 3Department of Environmental Science, Aarhus University, 4000 Roskilde, Denmark; martin.hansen@envs.au.dk; 4UFZ Department of Ecological Chemistry, Helmholtz Centre for Environmental Research, Permoserstraße 15, 04318 Leipzig, Germany; gerrit.schuurmann@ufz.de; 5Institute of Organic Chemistry, Technical University Bergakademie Freiberg, Leipziger Straße 29, 09596 Freiberg, Germany; 6Zebrafishlab, Department of Veterinary Sciences, University of Antwerp, 2610 Wilrijk, Belgium; dries.knapen@uantwerpen.be; 7AquaTT, Olympic House, Pleasants Street, D08 H67X Dublin, Ireland; marieke@aquatt.ie; 8Functional Genomics of Thyroid Hormone Signaling Group, Institut de Génomique Fonctionnelle de Lyon Ecole Normale Supérieure de Lyon, 69007 Lyon, France; Frederic.flamant@ens-lyon.fr; 9UMR7221 Molecular Physiology and Adaption, Centre National de le Recherche Scientifique—Muséum National d’Histoire Naturelle, CEDEX 05, 75231 Paris, France; laurent.sachs@mnhn.fr; 10Abt. Ökophysiologie und Aquakultur Leibniz-Institut für Gewässerökologie und Binnenfischerei, 12587 Berlin, Germany; werner.kloas@igb-berlin.de; 11RECETOX, Faculty of Science, Masaryk University, Kamenice 5, 625 00 Brno, Czech Republic; klara.hilscherova@recetox.muni.cz; 12Laboratoire Recherche Environnementale, L’ORÉAL Recherche & Innovation, 93601 Aulnay-sous-Bois, France; marc.leonard@rd.loreal.com; 13German Environment Agency (UBA), Section IV2.3 Chemicals, 06844 Dessau-Roßlau, Germany; Juergen.arning@uba.de; 14BASF SE, Experimental Toxicology and Ecotoxicology, 67098 Ludwigshafen, Germany; volker.strauss@basf.com; 15Graduate School of Nanobioscience, Yokohama City University, Yokohama 236-0027, Japan; taiseni@hotmail.co.jp; 16Centre for Organismal Studies, University of Heidelberg, 69120 Heidelberg, Germany; lisa.baumann@uni-heidelberg.de

**Keywords:** endocrine disruption, thyroid hormone disruption, AOP, adverse outcome pathway, OECD, test guideline, integrated approach to testing and assessment, IATA, cross-species extrapolation, biomarkers

## Abstract

ERGO (EndocRine Guideline Optimization) is the acronym of a European Union-funded research and innovation action, that aims to break down the wall between mammalian and non-mammalian vertebrate regulatory testing of endocrine disruptors (EDs), by identifying, developing and aligning thyroid-related biomarkers and endpoints (B/E) for the linkage of effects between vertebrate classes. To achieve this, an adverse outcome pathway (AOP) network covering various modes of thyroid hormone disruption (THD) in multiple vertebrate classes will be developed. The AOP development will be based on existing and new data from in vitro and in vivo experiments with fish, amphibians and mammals, using a battery of different THDs. This will provide the scientifically plausible and evidence-based foundation for the selection of B/E and assays in lower vertebrates, predictive of human health outcomes. These assays will be prioritized for validation at OECD (Organization for Economic Cooperation and Development) level. ERGO will re-think ED testing strategies from in silico methods to in vivo testing and develop, optimize and validate existing in vivo and early life-stage OECD guidelines, as well as new in vitro protocols for THD. This strategy will reduce requirements for animal testing by preventing duplication of testing in mammals and non-mammalian vertebrates and increase the screening capacity to enable more chemicals to be tested for ED properties.

## 1. Introduction

Recently, international workshops and projects arranged by the European Commission (EC) and OECD (Organization for Economic Cooperation and Development) have identified gaps in the testing of suspected endocrine disruptors (EDs) [1,2,3]. Regulators are therefore requesting better tests and approaches for the assessment of the hazards and risks of EDs, to protect human health and the environment. Based on workshop recommendations and identified gaps, the EC made a call under Horizon 2020 Research and Innovation Actions (H2020-SC1-BHC-2018-2020, Topic: SC1-BHC-27-2018: New testing and screening methods to identify endocrine disrupting chemicals) to support research enabling the gaps to be filled. ERGO (acronym for EndocRine Guideline Optimization) and 7 ‘sister’ projects signed consortium agreements with the EC in 2019 and formed the EURION cluster [4]. EURION facilitates collaboration and data sharing between the 8 projects and prevents research overlaps.

In ERGO, we address additional challenges identified by the EC and OECD. In both EU and other international legislations, regulatory procedures for the identification and assessment of EDs are separated for human health and the environment. Consequently, useful data obtained from non-mammalian vertebrate tests (e.g., fish and amphibians) are disregarded, or not given sufficient weight, in human health assessments and vice versa, even though the endocrine system is highly conserved among vertebrate classes [5,6,7,8,9,10,11]. The international workshops pointed out thyroid hormone disruption (THD) as a focus area, because existing vertebrate in vivo tests are not environmentally protective enough and validated in vitro tests for THD are not yet available.

THD has been linked with certain neurodevelopmental disorders, such as autism, attention-deficit/hyperactivity disorder (ADHD), learning disabilities and mental retardation, including decreased IQ and modified brain structure in children [12,13]. These disorders have been reported to increase over the last four decades [14,15]. Meanwhile, wildlife and human contamination by anthropogenic chemicals, including EDs, is well documented [16,17], and recommendations have been made to more systematically evaluate the potential developmental toxicity of new and existing chemicals [14]. Attempts have been made to evaluate the societal cost of human and environmental exposure to EDs, which have been controversially discussed in the scientific community [18,19,20]. In view of the increasing emotional concerns in the media and public, the exposure to, and the resulting impact of, EDs must be properly discussed under strict scientific standards.

Nonetheless, as reported by [21], even EU REACH (Registration, Evaluation, Authorization and Restriction of Chemicals) dossiers of high tonnage chemicals (above 100 and 1000 tonnes/year) are lacking measured developmental toxicity data. Often, animal tests have been waived through read-across and the exploitation of existing data, but not by employing alternative test methods [22]. Additionally, ED-related endpoints and effects regarding the environment are still missing in REACH standard data requirements. With respect to this, it is currently discussed to amend the REACH standard data sets, to explicitly cover the ED mode of actions and correlated adverse effects. To efficiently cover ED in the tiered REACH testing approach, it is of utmost importance to use human health data for environmental assessment and vice versa. According to the OECD conceptual framework (CF) [23], only in vivo animal tests are able to identify adverse effects induced by ED pathways. In addition to the ethical aspects of using animals, such in vivo tests are lengthy and expensive. Moreover, testing on animals (as defined by Dir. 2010/63/UE) is not allowed in Europe (and some other countries) for the safety assessment of cosmetic ingredients and low tonnage chemicals (less than 10 tonnes/year), which must rely on alternative methods. Consequently, these compounds can only be tested with OECD conceptual framework (CF) level 2 (in vitro) and some level 3 tests (fish and amphibian early life stages). The latter provide data about endocrine mechanisms, but are currently not designed to detect population-relevant adverse effects (AE).

ERGO is a coordinated attempt to help fill the gaps in the field of THD for human health and the environment. It will allow the identification of both disturbance of the thyroid axis and its potential adverse effects in different vertebrate classes, including humans. ERGO is expected to improve methodologies for using cell tests and fish and amphibian assays for the early screening of substances. It will also develop new in silico models for predicting the internal dose of THDs to design physiologically based toxicokinetic (PBTK) models and to link molecular initiating events (MIEs) with AE, within an adverse outcome pathway (AOP) network. PBTK models enable quantitative descriptions of absorption, distribution, metabolism, and the excretion of chemicals in biota, and inform about how compound properties and physiological characteristics affect the chemical’s fate in the organism [24].

ERGO is expected to increase basic knowledge on the detailed effects of TH disturbances. The ERGO approach will be of significant interest for the safety assessment of existing chemicals lacking endocrine and developmental toxicity data and new chemicals at an early stage of their industrial development. This methodology should allow:The simultaneous screening of chemicals for their potential human THD effects, as well as their environmental impact with similar adverse effects on e.g., fish and amphibians.Significantly reducing the requirement for vertebrate animal testing, thus complying with the 3Rs principle.Assessment at the vitro scale providing new clues for automation and higher throughput screening of chemicals, which would further reduce the cost of their assessment.

## 2. Concept

ERGO will provide evidence that bridging mammalian and non-mammalian testing for the identification of EDs is justified for chemicals affecting endocrine axes across vertebrate classes. The proof of concept of cross-vertebrate extrapolation of ED effects will be presented for the thyroid system (Figure 1). In current relevant regulations, both in the EU and worldwide, the identification of EDs and the follow-up in terms of hazard and risk assessment is separated between mammalian and non-mammalian testing. The separation is historically based on general systemic toxicity testing, where best practice has been, and still is, species- and class-specific assessment, because systemic toxicity is recognized to be highly dependent on species- and class-specific absorption, distribution, metabolism and excretion (ADME).

The existing separation in ED testing between mammals and non-mammals is clearly outlined in the OECD Conceptual Framework for Testing and Assessment of Endocrine Disrupters, where all in vivo testing is divided into mammalian and non-mammalian toxicology test guidelines [23]. There is, however, increasing scientific and regulatory awareness of the value of extrapolating data between vertebrate classes, and this is highlighted in several sections of the 2018 update of OECD guidance document 150 [25]. Strong and increasing scientific evidence supports read-across between mammals, fish and amphibians, and thus, ERGO will establish test systems with lower vertebrates (fish and amphibians) for early warning and screening purposes, for not only environmental, but also human health. To this end, a detailed understanding of the mechanisms underlying the interference of THDs with the highly conserved vertebrate thyroid system is required to allow the comparison of results across vertebrate classes [26,27]. Teleost fish, the phylogenetically oldest and largest group of vertebrates, will be used to achieve this goal. Specifically, zebrafish (*Danio rerio*), the most popular fish model for toxicological and vertebrate developmental research, is being used. A strong basis of information on the regulation of specific developmental processes by THs [28,29] has made zebrafish an important biomedical model for TH-related diseases, including obesity, cardiovascular diseases and diabetes [30]. Thus, a comprehensive database on structure and function of the zebrafish thyroid system is available; the ontogenetic patterns of thyroid receptors (TR) L-TRαL-Thraa, S-Thraa, Thrab and Thrb are fully described [26,31,32], and the function of the zebrafish thyroid system appears to be fully comparable to that of higher vertebrates and humans. For instance, the molecular mechanisms of thyroid organogenesis and the role of TR signaling during embryogenesis are well documented for zebrafish and seem to be highly conserved between zebrafish and mammalian models [26,31]. (Neuro)developmental processes that are, at least partially, regulated by THs include eye development, swim bladder inflation and fin formation during early embryonic development and the metamorphosis from the larval to juvenile stage [33,34,35,36,37]. Exposure to THDs has been shown to affect these important developmental processes, which makes them promising candidates for thyroid-related endpoints in THD fish testing assays. For instance, inhibited differentiation of paired fins was observed in zebrafish exposed to different THDs during the transition from larval to juvenile stage [37]. Moreover, eye development and the resulting visual performance were demonstrated to be disturbed by disruption of the thyroid system with different THDs or thyroid-specific gene-knockouts [33,36,38,39,40,41]. For the impact of THDs on the swim bladder inflation of zebrafish, a putative AOP network already exists (Figure 2, AOP nrs. 155–159, http://aopwiki.org), which will be further used for the potential implementation of this endpoint in fish THD testing. The use of zebrafish for screening for general toxicity and ED effects is common practice, as evidenced by multiple OECD test guidelines (TGs) (e.g., 210, 229, 230, 234, 236). However, none of these test systems include biomarkers/endpoints (B/E) addressing effects on the (hypothalamus pituitary thyroid) HPT axis, which is one of the specific goals of ERGO. The inclusion of thyroid endpoints in OECD fish TGs has recently been added as a project to the OECD Work Plan for the Test Guidelines Programme (Project 2.64), and the AOP network on thyroperoxidase and/or deiodinase inhibition, leading to impaired swim bladder inflation in fish during early life stages is part of the OECD development program workplan (Project 1.35). We therefore expect that, after further development of the AOP network, the addition of thyroid-related AOP-supported endpoints to specific OECD TGs will be achievable within the timeframe of ERGO.

Together with already existing literature data, ERGO will provide a comprehensive data set to demonstrate that fish, i.e., zebrafish, are a fully adequate and effective model to assess THD effects and extrapolate the results to other vertebrate classes including humans. Similar data sets are currently being generated for amphibians, as over many years, the amphibian model organism *Xenopus*
*(laevis and tropicalis)* has contributed fundamentally to research in biomedicine, neurobiology, physiology, molecular biology, cell biology, and developmental biology, making it one of the best investigated animal models [42,43]. The frog model has provided major insight into in vivo mechanisms of TH signaling [44], since TH-dependent developmental changes like metamorphosis are directly observable and quantifiable. The extreme sensitivity and responsiveness of amphibian metamorphosis to TH signaling is exploited in OECD TGs 231, 241 and 248.

Both at the molecular and morphological levels, metamorphosis in lower vertebrates bears strong similarities with perinatal postembryonic development in mammals. In general, TH signaling including TH transport across cell membranes, metabolism by deiodinases, and molecular mechanisms of gene regulation (TH receptors, transcriptional cofactors, and chromatin remodeling) are conserved to a high degree in humans, fish and amphibians (Figure 3) [27].

As outlined above, both EC and OECD have set a high priority in developing new approaches for the predictive ED assessment fit for regulatory purposes. In this context, THD is known as key challenge because of lacking information-rich in vitro approaches, as well as in vitro to in vivo and environmental-human extrapolations.

ERGO was built in response to these needs and a respective EU H2020 call (see above). In the ERGO consortium, there are several partners with long-lasting expertise with work on the endocrine system of different vertebrates. As described above, evidence in the peer-reviewed literature supports the conservation of large parts of the vertebrate endocrine system, and this evidence is growing fast, particularly because of a focus on AOP development and increasing knowledge about ED modes of actions (MOA), including in silico modelling, biotransformation and transcriptomics data. This proof of conservation of the endocrine system on the one hand and the separated testing strategies on the other was the basic motivation for ERGO to develop the AOP network based on a cross-vertebrate class effects approach. The thyroid system was selected due to reasons already outlined, however cross-talk investigations with other conserved endocrine axes like the hypothalamus pituitary gonadal (HPG) axis could be included in the cross-class approach in the future as well [45].

## 3. Approach

The ERGO project was launched in January 2019 and consists of 15 partners from academia, industry and regulatory bodies. ERGO is divided into eight work packages (WPs), supporting an efficient research and outreach structure (see below).

An initial joint effort has been made to select
MIEs subject to ERGO in vivo researchMIEs subject to ERGO in vitro researchIn vivo and in vitro ERGO reference compounds

In addition to covering MIEs of primary THD relevance, the bioassay feasibility and the chemical domain (see below) of the test compounds were taken into account. The resulting set of 28 ERGO reference compounds covers the following MIEs: thyroid hormone receptor (THR), thyroid transport protein (TTR), thyroid peroxidase (TPO), sodium-iodide-symporter (NIS), deiodinases I-III (DIO), and thyroid binding globulin (TBG). Besides in vitro profiling of all ERGO reference compounds, the following subset of six compounds has been selected for generating in vivo reference data: ampicillin, carbamazepine, iopanoic acid, perchlorate, propylthiouracil and tetrabromobisphenol A (see WP5). In the following, the WP implementation of the ERGO research work in the eight WPs is outlined.

### 3.1. WP1 Coordination

Besides an ERGO-internal Project Office responsible for the overall management, WP1 coordinates our interaction with an international Scientific Advisory Board (SAB), with members from academia and regulatory bodies from EU, the UK, North and South America. The SAB gives advice on scientific issues and disseminate ERGO concepts and results to other communities and regulatory bodies. WP1 represents ERGO in the OECD, including the OECD Validation Management Group for Ecotoxicity Testing (VMG-Eco) and OECD Validation Management Group for Non-Animal Testing (VMG-NA). VMG-Eco has the role of discussing and validating new and updated TGs and biomarkers for the environment. Partners in ERGO are Co-chairs of VMG-Eco and responsible for OECD TG project 2.64; “Inclusion of thyroid endpoints in OECD fish Test Guidelines”.

### 3.2. WP2 Knowledge Management and Data Basing

To integrate the work performed within the different WPs and to align the scientific tasks with the overarching strategy of the ERGO project, structured and efficient data management is crucial. A searchable and well-structured electronic database (Figure 4), containing test data as well as literature data and relevant meta information, supports decision making on relevant project issues, like the selection of adequate case study chemicals at the beginning of the project. Additionally, it is essential for the success of the project to involve all interested stakeholders from the beginning to integrate their knowledge and needs. This is done by establishing a user reference group compiled of experts from industry, regulatory bodies and scientists.

### 3.3. WP3 Adverse Outcome Pathway (AOP) Network Development

Based on comparative data compilation and comprehensive scientific evidence, an AOP network covering multiple modes of THD will be developed. Structuring all available scientific evidence according to formal OECD AOP development principles and following the OECD review process will support the prioritization of assays for identifying THDs across vertebrate classes for validation. Both experimental data from literature (WP2) and new data generated by the consortium (WP4, 5, 6) are used. As a first step, existing data on the TH regulation of eye development of fish have been summarized and a new, hypothesized AOP network is currently being developed and regularly updated with new data generated in ERGO experiments. Evidence will be added to existing AOPs (see Figure 2) and new AOPs will be developed if gaps exist, e.g., interference with thyroid serum-binding protein as a MIE. WP3 will amend the relevant AOPs in the AOP-Wiki to promote international collaboration and visibility. Due to the principle of re-usability of KEs included in individual AOPs, this will de facto lead to the construction of an AOP network in the AOP-Wiki.

In addition, ERGO leads the AOP working group (WG) of the EURION cluster. There are currently around 50 individual WG members, with representatives from each of the eight cluster projects. The mission of the AOP WG is twofold: to support and facilitate AOP-related activities in the EURION projects, and to bring together AOP-structured information and data across projects. The WG is organizing AOP training workshops as a function of needs within the projects, AOP development workshops as data becomes available, and regular meetings and teleconferences to align AOP-related activities and stimulate collaboration across projects. Examples of specific opportunities for collaborative AOP development that have been identified include THD (ATHENA, ERGO, SCREENED and ENDpoiNTs) and developmental neurotoxicity (ATHENA and ENDpoiNTs).

### 3.4. WP4 Modelling and Biotransformation

This WP addresses the chemical perspective of THD, building on existing literature and recent research projects [46,47,48]. The focus is on how chemical structure triggers THD MIEs and MOAs within and across species, and how this can be predicted in silico, taking into account metabolic half-lives and biotransformation pathways of reference substrates, and correspondingly informed physiologically based toxicokinetic modelling (PBTK). The major deliverable will be an in silico decision support system (DSS) for the predictive THD potential of chemical substances, featuring qualitative and quantitative structure-activity relationships (QSARs), consensus modelling and Bayesian statistics, to account for conflicting input information, as well as varying levels of information and confidence.

The initial focal points were to populate our database with literature-known TH disruptors, to characterize the chemical domain of our ERGO reference compounds as related to the EINECS inventory (see below), to undertake computational docking with reference ligands, and to profile biotransformation through S9 enzyme assays derived from rat hepatocytes (such as the S9 mix used for the Ames test). Respective biotransformation studies include a fish-specific S9 enzyme assay (OECD 319B) [49] and the results feed the PBTK modelling regarding metabolism.

### 3.5. WP5 Case Studies for Thyroid-Related Endpoints and Biomarkers in ED Test Systems

By far, the largest and most experimental WP in ERGO. Based on an analysis of gaps in the database on effects of THDs in vertebrates, ERGO WP5 will (1) identify suitable B/E for THD across several vertebrate classes and, thus, (2) provide the experimental data required to implement THD as an endpoint in existing or novel OECD TGs. Figure 5 outlines a potential case study with chemical X in WP5. Such case studies are performed with a battery of THDs, with six different defined MOAs (Appendix A): (1) NIS inhibition, (2) THR interaction, (3) TPO inhibition, (4) DIO inhibition, (5) TTR/TBG interaction, (6) TH liver clearance. For each MOA, different model compounds have been selected based on available information from literature, which provides sufficient evidence for THD-specific MOAs of these compounds, without undesired side effects. The selected compounds are: ampicillin (negative control), carbamazepine, iopanoic acid, perchlorate, propylthiouracil and tetrabromobisphenol A. Each compound is tested and compared across all test systems, with different vertebrate classes and life stages. B/E are being assessed according to the AOP concept, ranging from the molecular, over morphological, up to the physiological level. Special emphasis is put on the identification of new THD-related B/E in fish, preferably in embryonic stages to comply with the 3R principle. Current focus of the experimental work in WP5 is on THD-induced disturbance of the development of eyes and swim bladder in fish. The data basis for the latter is very broad, and a putative AOP network has already been developed (see Figure 2). A similar data set on the effects THDs on eye development is currently being collected from literature and experiments performed by different ERGO partners. The first results show that the eye development could serve as a meaningful B/E in already existing OECD TGs with fish, including early life stages. These new fish B/E are compared with already established B/E from amphibian testing and potentially new B/E for amphibians are assessed as well.

An in vitro bioassay battery is being set up to address identified cross-species priority molecular initiating/key events and to support in vivo studies. The models for studying the prioritized endpoints and a set of biomarkers include human/mammalian cell lines from thyroid, liver and neural stem cells. The effect of 2D and 3D cultivation of the cells on TH balance relevant endpoints is examined. The in vitro bioassay battery includes the assays listed in Figure 5.

### 3.6. WP6 Mammalian Endpoints and Epidemiology

Current mammalian assays for THDs have been criticized for both ethical and technical reasons. In compliance with the 3Rs principle, WP6 proposes to refine these assays to improve sensitivity and precision, and thus to reduce the number of animals notably by increasing the precision of the endpoint’s measurement. A distinction is made between endpoints which relate to the maintenance of steady state levels of T4 and T3 in blood, and endpoints which represent the tissue response to T3. These deserve special attention, because THDs can alter neurodevelopment and metabolism without modifying the circulating levels of T4/T3. WP6 explores the possibilities currently offered by “omics”, combining metabolomics (mass spectrometry), transcriptome analysis (RNAseq), and the genome-wide analysis of chromatin occupancy (ChipSeq) to characterize T3 signaling and the AO of THD exposure. A new transgenic mouse model (Cre:LoxP technology) to restrict the expression of T3 receptors with mutation or tags [50] allows the study of T3 signaling in specific cell types with intact tissues. An epidemiological study including the CELSPAC birth cohort with focus on a potential role of exposure in THD-related developmental disorders will employ an AOP network-based strategy and will be closely coordinated with activities within the European Human Biomonitoring Initiative (HBM4EU).

### 3.7. WP7 Pre-Validation and Validation of Biomarkers and Endpoints

The most valuable tools in the regulation of chemicals are standardized TGs with sensitive B/E, that address the concerns of specific hazardous properties of chemicals. Such test methods have been developed for the evaluation of EDs in the regime of OECD, to ensure international standardization and mutual acceptance of data. WP7 will support OECD TGs with new B/E for THD. B/E selected, developed and tested in WP2,3,4,5 and 6 will only go into WP7 for validation or pre-validation if they have been evaluated as suitable for the cross-class extrapolation of effects from non-mammalian vertebrates to mammals. Dependent on the TRL (technological readiness level) of the B/E, either a pre-validation will be performed within ERGO or an OECD validation including external participants will be carried out.

### 3.8. WP8 Dissemination and Exploitation

WP8 ensures that the outcomes and achievements of ERGO are effectively transferred to target- and end-users and that there is a measurable impact of ERGOs results on society. A dissemination and exploitation plan (DEP) is integrated in the project design, in order to maximize exposure of the project’s progression and contributions to science and society. A tailored knowledge transfer methodology is used to identify potential scientific, industrial and societal applications of the project’s outputs across a variety of sectors and ensure they achieve measurable impact. In coordination with the Innovation Task Force and WP2, WP8 has established supporting protocols for the management of Intellectual Property of ERGO’s results. ERGO has established a dedicated project website (www.ergo-project.eu) as the main tool for promoting the project and disseminating the project’s objectives, work plan and results to a wide audience, including all stakeholders and possible end-users. To ensure successful promotion of the project and to sustain the interest of the target audience and attract new users, the website’s content will be maintained, continuously updated and populated with new information throughout the project’s lifetime. The website will remain active for five years after the end of the project, to serve as a valuable public resource of research information on the subject and for promoting the outputs of publicly funded research in the domain beyond the project’s lifetime. Social networking is part of the ERGO communication strategy and a dedicated project Twitter account (@ERGO_EU) was set up at the start of the project and is used to publicly “tweet” ERGO relevant information. Project-related tweets are posted regularly in accordance with the H2020 Programme Guidance Social media guide for EU funded R+I projects.

In addition, ERGO leads the EURION Communication working group and have set up and are maintaining the cluster website and social media activities.

## 4. ERGO vs. EINECS Chemical Domain

An important issue regarding in vitro bioassays are factors controlling the bioavailability of test substances. In this context, compound dissolution may be hampered by sorption (to vials as well as to extracellular matrix) and volatilization, which in turn are driven by hydrophobicity and Henry’s law constant, respectively. Moreover, these physicochemical properties affect both toxicokinetics and toxicodynamics, considering the variation in compound affinity for water-rich vs. water-poor tissues (compartments), pathways and receptor sites. From this viewpoint, it is of interest to profile the physicochemical space covered by our ERGO reference compounds.

According to EPISuite [51] and ACD/Percepta [52] calculations (Figure 6), the approximate property ranges of the ERGO reference set of compounds (Appendix A) are as follows: log *K*_ow_ (hydrophobicity, octanol-water partition coefficient) from −5 to 8, log *H* (Henry’s law constant, [Pa m^3^ mol^−1^]) from −17 to 1, log *S*_w_ (water solubility, [mol L^−1^]) from −11 to 1), and p*K*_a_ (dissociation constant), indicating the predominant speciation at pH 7. This wide physicochemical profile addresses potential in vitro challenges regarding sorption, volatilization and resultant bioavailability.

In Figure 7, the ERGO physicochemical space (red) is compared with the one of the EINECS subset of 56703 compounds (grey), that have well-defined chemical structures and yield formally valid EPISuite results. In this graph, the calculated properties of EINECS compounds outside reasonable ranges (as specified in the right part of the figure) are collected at the respective property edges (light grey), facilitating visual inspection of the ERGO vs. EINECS setting. The latter shows that the 28 ERGO reference substances cover a significant portion of the EINECS domain regarding bioavailability-related properties.

Although the physiochemical domain provides an important aspect of a given compound set, it does not inform directly about the associated structural domain. For the latter, the atom-centered fragment (ACF) approach [53,54] has been applied for the ERGO-EINECS comparison, considering all 72,520 EINECS compounds with defined chemical structures. After correcting for various structural chemistry issues in EINECS, the result is as follows: one of our 28 ERGO reference compounds (DON) is borderline outside, one is borderline inside (BDE47), three (PTU, MMI, HBCD) are inside EINECS with however quite unique structural features, and 23 (82%) have structural features well covered by EINECS. This probably reflects the fact that bioactive compounds may contain structural features which are systematically different from industrial compounds. Moreover, it demonstrates that substances with physiochemical properties fitting well to a certain chemical inventory (here: EINECS) may nevertheless be outside the respective structural domain. It follows further that the ERGO reference compounds cover both EINECS-typical and outside-EINECS structural features and highlights the importance in addressing chemical structure when assessing chemical domain belongings.

## 5. Expected Impact

ERGO is expected to develop an IATA across mammalian (rodent) and non-mammalian (fish, amphibian) vertebrates as proof-of-concept, to assess compounds for their THD potential through a battery of AOP-targeted in vivo experiments, in vitro bioassays, omics and in silico information. To this end, novel MOA-specific B/E and structure-activity relationships will be profiled for their predictive THD information content, with particular attention for within-species vs. across-species AOP networks and their MIEs and KEs. In particular, state-of-the-art molecular biology in combination with in vitro research and its assessment through judiciously selected reference investigations in vivo will be complemented by in silico information of molecular-level THD action regarding protein-ligand interaction and PBTK. Accordingly, major outcomes include:Identification of novel THD-related endpoints for in vivo testing with non-mammalian vertebratesMIE/KE-specific in vitro protocols to assess the potential of THDs for triggering respective AOPsMIE/KE-specific omics profiles informing about AOP progressOpportunities for AOP information across mammalian and non-mammalian vertebrate TH disruptionTHD-relevant AOP crosstalk patterns, andStructure-activity insight into AOP/THD-specific MIEs

These will serve as novel components of an overall IATA, augmented by a decision-support scheme in order to help converting many-endpoint/method information into a coherent THD assessment strategy. The respective proof-of-concept is envisaged to be ready for subsequent OECD validation, thus fostering a refinement, reduction and replacement of in vivo THD testing, in line with current needs to overcome practical 3R barriers in the REACH context. Furthermore, the results obtained here can be used to support conclusions on ED properties of substances, e.g., under the biocidal and plant protection framework without the need of extensive in vivo testing.

## 6. Outreach and Implementation

During the course of the project, ERGO will ensure close collaboration and harmonization with other European projects and initiatives on ED testing, e.g., the ongoing JRC work on THD in vitro testing or different EU Tender projects on optimization of existing OECD TGs. Based on the outcomes of the knowledge management and transfer activities, ERGO will organize three focused workshops that will aim to transfer project knowledge to priority target users. The first workshop will be targeted at end users of OECD TGs, including contract laboratories and larger enterprises, intended to share the results of ERGO and feed them into policy and regulatory processes. A second workshop will be opened up to a broader range of actors involved in the regulatory process across Europe, sharing ERGO knowledge outputs such as scientific findings, recommendations, and datasets, that could be taken up and applied by others. A specific training session will be provided on how to use key outputs. The third workshop will be a final project showcase event to share the achievements of the project to a wide stakeholder base interested in hearing about the results and impacts of ERGO. Representatives from EURION and any other projects and initiatives working on EDs will be invited to these events. At the end of the project, a key achievements publication outlining the knowledge outputs generated by ERGO, and the transfer activities that took place within the project, will be widely distributed for outreach to wider society. It will include a roadmap for post project actions that may be needed to maximize the impacts of the project.

## 7. Conclusions

The overall concept of ERGO is to improve the hazard and risk assessment of EDs for the protection of human health and the environment, by introducing a paradigm shift in the scientific and regulatory use of TG data. ERGO aims to break down the existing wall between mammalian and non-mammalian vertebrate testing, by demonstrating that it is feasible to extrapolate effects of EDs across vertebrate classes, i.e., an adverse effect on an endocrine-specific endpoint observed in a fish or amphibian study will also raise concerns about a possible adverse effect in humans. Thyroid-related biomarkers and apical endpoints suitable for the extrapolation of effects in fish and amphibians to humans and other mammals (and vice versa) are investigated, evaluated and finally validated for inclusion in existing or new OECD TGs. A cross-class AOP network will provide the scientifically plausible and evidence-based foundation for the selection of B/E in lower vertebrate assays predictive of human health outcomes. These assays will be prioritized in ERGO, in preparation for international validation through OECD. ERGO will also develop and implement a novel concept of an AOP network-based strategy for epidemiological and human exposure studies (e.g., from the workplace), allowing the evaluation of mechanistically based associations of external and internal exposure with THD-related health disorders. This strategy enables the prioritization of risk drivers and provides information for the improved hazard and risk assessment of THD.

## Figures and Tables

**Figure 1 ijms-21-02954-f001:**
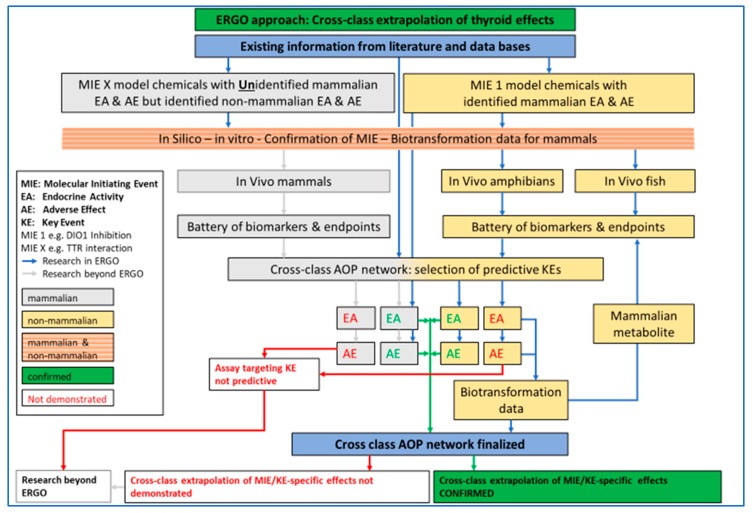
ERGO (acronym for EndocRine Guideline Optimization) approach for testing and cross-class extrapolation of THD (thyroid hormone disruption) effects between mammalian and non-mammalian test guidelines. Based on existing information, chemicals with known molecular initiating events (MIE), key event (KE), endocrine activity (EA) and adverse effect (AE) in mammals will be investigated in silico, in vitro and in vivo, in fish and amphibians. A battery of biomarkers and endpoints (B/E) will be tested for EA and AE confirmation in the in vivo studies. Biotransformation data will be obtained when EA and/or AE cannot be confirmed. All existing and new data will be used for AOP (adverse outcome pathway) network development. Chemicals with unknown effects in mammals but known EA and AE in non-mammalian vertebrates will be investigated in silico, in vitro and for differences in biotransformation among mammals and non-mammalian vertebrates. Blue arrow: Research in ERGO, Grey arrow: Research beyond ERGO, Green arrow: confirmed cross-class extrapolation, Red arrow: cross-class extrapolation not demonstrated.

**Figure 2 ijms-21-02954-f002:**
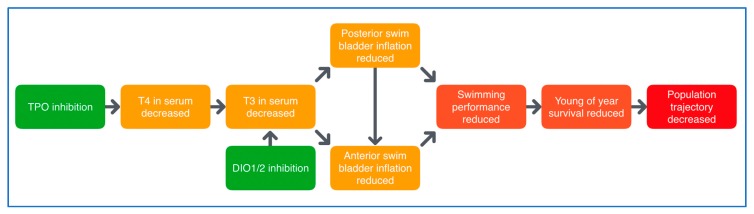
Putative AOP network for deiodinase (DIO) and thyroperoxidase (TPO) inhibition, leading to impaired swim bladder inflation in fish. DIO inhibition causes triiodothyronine (T3) serum decrease and reduced anterior swim bladder inflation, with reduced swimming performance and survival as adverse consequences. https://aopwiki.org/aops/155-159.

**Figure 3 ijms-21-02954-f003:**
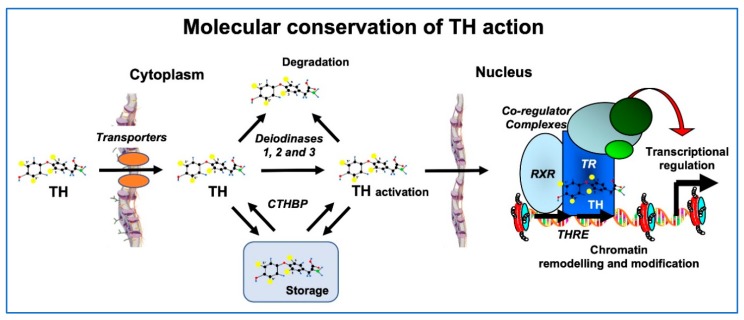
Conservation of TH signaling in vertebrates. Processes from TH transport into the cell to control of gene expression share homologous proteins and mechanisms in humans, fish and amphibians. Homologous TH transporters enable TH entry into cells, where deiodinases function to activate or deactivate it. Cytoplasmic TH binding proteins (CTHBPs) modulate cytoplasmic availability. In the nucleus, the TH receptor (TR) forms a heterodimer with the retinoid X receptor (RXR) and binds to DNA at TH response elements (THRE), where co-regulator complexes are recruited to alter the state of chromatin, ultimately leading to induced expression of TH response genes. Modified from [27].

**Figure 4 ijms-21-02954-f004:**
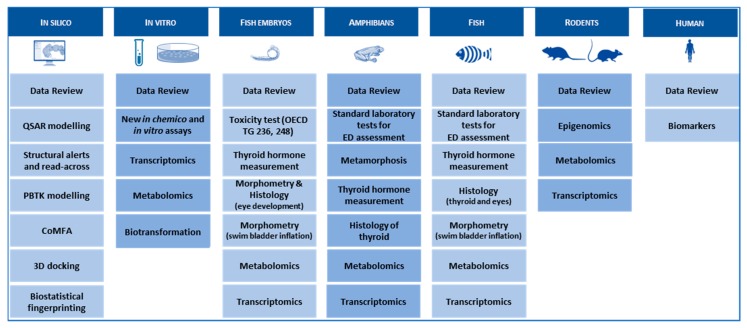
ERGO data warehouse. Via a combination of existing data from data reviews and data generated in ERGO WPs, ERGO will develop a comprehensive data warehouse composed of different kinds of cross-vertebrate class, in silico, in vitro, in vivo samples and cell-lines samples combined with many state-of-the-art analytical methods.

**Figure 5 ijms-21-02954-f005:**
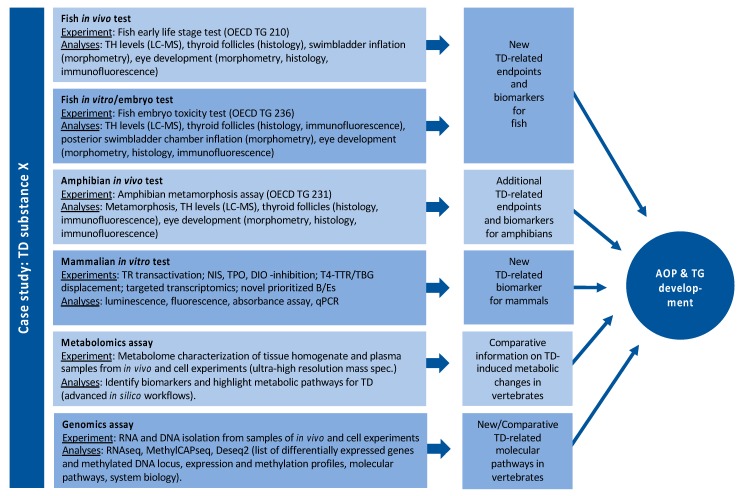
Example of a potential case study in WP5 with compound X.

**Figure 6 ijms-21-02954-f006:**
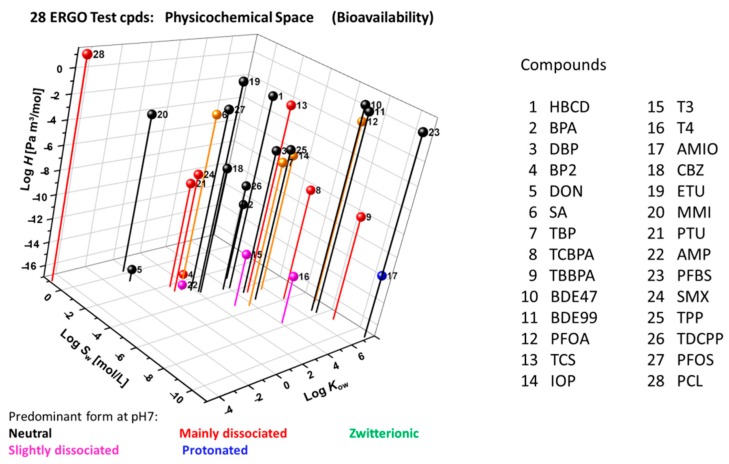
Physicochemical domain of 28 ERGO reference compounds in terms of log *K*_ow_ (octanol-water partition coefficient), log *H* (Henry’s law constant) and log *S*_w_ (water solubility) as calculated through EPISuite ([51], augmented by calculated p*K*_a_-driven [52] main speciation.

**Figure 7 ijms-21-02954-f007:**
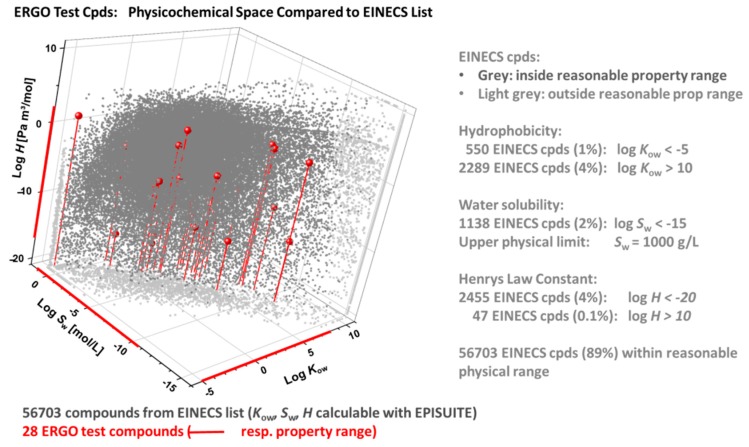
ERGO vs. EINECS physicochemical domain in terms of log *K*_ow_ (octanol-water partition coefficient), log *H* (Henry’s law constant) and log *S*_w_ (water solubility), as calculated through EPISuite [51]. Among 100,102 EINECS entries, 72,520 have defined chemical structures, of which 56,703 compounds with formally valid EIPSuite results (grey) are compared to the 28 ERGO reference compounds (red). EINECS compounds with calculated properties outside reasonable ranges are plotted in the property edges (light grey).

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
