# Peer review of "ERGO: Breaking Down the Wall between Human Health and Environmental Testing of Endocrine Disrupters"

_ijms, 2020, doi:10.3390/ijms21082954_

Round 1

Reviewer 1 Report

Reviewer’s comments and suggestions for Authors

The current project report by Henrik et al investigates the working aim of ERGO which is a European Union-Funded Research and Innovation action that discusses regular testing of endocrine disrupters in mammalian and non-mammalian vertebrate and to identify, to develop and alignment of thyroid-related biomarkers, their endpoints for linkage between the classes of a vertebrate. For the analysis, the study used an adverse outcome pathway network, which covers the several modes of thyroid hormone disrupter (THD) in various vertebrates. 

This report is going to provide the scientific evidence-based foundation for the selection of endpoints B/E and testings in lower vertebrates predictive of human health outcomes and thereby ERGO will reassess the ED testing strategies from in silico methods to in vivo testing and finally develop, optimize and validate existing in vivo and further develop the guidelines of OECD and protocol for THD.

The manuscript is in good shape but there was missing information in some paragraphs and that is why author need to incorporate the matter. Additionally, there are few points needed to revise in MS

(1) In the abstract line number 41, the OECD needs to be in full form.

(2) Line 62-63, please briefly discuss the lines and clear out the exact meaning in the paragraph.

(3) Line 93, what is CF level 2, it is better to write the full form here and what was the meaning of that, please make consistency about this in the whole MS 

(4) Line number 101, physiologically based toxicokinetic (PBTK) describe here in detail about the model

(5) In the concept section, only THYROID system is used only, why not others

(6) Figure 1 legend should be properly written 

(7) Line 154, why the neuro is in the bracket

(8) Line 157-160, the presented line needs references.

(9) Line 166 and 167, please write the full form of TG and HPT axis here only

(10) Line 211, No reason has been discussed earlier. It is better to simplify the paper with lots of description. The figures used in the paper have not been fully described.

Comments on approach section, Is the project related to THD. I am not convinced with the matter they presented here, the approach should be clear about using WP1 COORDINATION and the use of such systems.

(11) Figure 4, need to rewrite the title.

(12) Line 280, How the author discusses the biotransformation. they need to simply the paragraph and discuss the modelling used in this

(13) Line 292, why the only Thyroid related if the paper is specific to TRE then it is better to change the title so that the study could be more specific 

Line 367, It is better to discuss more on this section as it does not qualify to understand every aspect used in this part related to the physicochemical domain

(14) Expected impact, I think it is better to write insilico first then goes on invitro and invivo

(15) Please check some of the references have not been according to the journal guideline, follow the journal guideline. 18,31, 33 and many more

Author Response

Rebuttal letter

Reviewer #1

(1) In the abstract line number 41, the OECD needs to be in full form.

Reply: has been changed accordingly

(2) Line 62-63, please briefly discuss the lines and clear out the exact meaning in the paragraph.

Reply: thanks for pointing that out. We have separated and shortened the sentences for better readability.

(3) Line 93, what is CF level 2, it is better to write the full form here and what was the meaning of that, please make consistency about this in the whole MS 

Reply: has been changed accordingly

(4) Line number 101, physiologically based toxicokinetic (PBTK) describe here in detail about the model

Reply: thanks for addressing this issue. We added an explanation and a reference: “PBTK models enable quantitative descriptions of absorption, distribution, metabolism, and excretion of chemicals in biota, and inform about how compound properties and physiological characteristics affect the chemical’s fate in the organism”.

(5) In the concept section, only THYROID system is used only, why not others

Reply: we agree that it would be interesting to apply our project concept also to other hormonal axes. However, for the conceptualization of this 5-year project, we had to decide on a specific focus that is promising for our overall goal. As described in the introduction and the concept paragraphs, the TH system is highly conserved among vertebrates, and the lack of appropriate testing guidelines for this endocrine modality has been identified as important gap by the EU and OECD. However, we are convinced that our approach will also be applicable to other hormonal axes, as outlined in lines 210-213: “The thyroid system was selected due the reasons already outlined, however cross-talk investigations with other conserved endocrine axes like the Hypothalamus Pituitary Gonadal (HPG)-axis could as well be included in the cross-class approach in the future [44]”.

(6) Figure 1 legend should be properly written 

Reply: We have augmented the legend by explanations of the abbreviations (that were already defined in the previous text).

(7) Line 154, why the neuro is in the bracket

Reply: because the processes we describe in the following are only partially related to neurodevelopment (eye development), the others are part of general development. So instead of writing both words, we chose this common way of combining a specific with a general term.

(8) Line 157-160, the presented line needs references.

Reply: thanks for pointing this out. The references were given in line 162, but we have now distributed them among the specific sentences.

(9) Line 166 and 167, please write the full form of TG and HPT axis here only

Reply: has been changed accordingly

(10) Line 211, No reason has been discussed earlier. It is better to simplify the paper with lots of description. The figures used in the paper have not been fully described.

Reply: Partly addressed. We have added a (further) short motivation about why ERGO had been set up. Other than that, we are not sure about what the reviewer actually meant with “no reason”. By contrast, we believe that our introduction provides a comprehensive justification of why ED in general and THD in particular are key areas with a high need for developing new approaches fit for regulatory purposes, which was the reason to build ERGO as respectively EC-funded research project. Moreover, all figures are shortly described and referred to in the text.

Comments on approach section, Is the project related to THD. I am not convinced with the matter they presented here, the approach should be clear about using WP1 COORDINATION and the use of such systems.

Reply: Addressed. We have shortened and modified the WP1 description, now focusing on features specific for ERGO:

“Besides an ERGO-internal Project Office responsible for the overall management, WP1 coordinates our interaction with an international Scientific Advisory Board (SAB) with members from academia and regulatory bodies from EU, the UK, North- and South America. The SAB gives advice on scientific issues and disseminate ERGO concepts and results to other communities and regulatory bodies. WP1 represents ERGO in the OECD including the OECD Validation Management Group for Ecotoxicity Testing (VMG-Eco) and OECD Validation Management Group for Non-Animal Testing (VMG-NA). VMG-Eco has the role of discussing and validating new and updated TGs and biomarkers for the environment. Partners in ERGO are Co-chairs of VMG-Eco and responsible for OECD TG project 2.64; “Inclusion of thyroid endpoints in OECD fish Test Guidelines”.

(11) Figure 4, need to rewrite the title.

Reply: thanks for pointing this out. We added the title “ERGO data warehouse”.

(12) Line 280, How the author discusses the biotransformation. they need to simply the paragraph and discuss the modelling used in this

Reply: Addressed. We have added further information about the S9 enzyme assay and its role to inform the PBTK modelling about metabolic half-lives and pathways:

“..taking into account metabolic half-lives and biotransformation pathways of reference substrates, and correspondingly informed physiologically based toxicokinetic modelling (PBTK).”

“..and to profile biotransformation through S9 enzyme assays derived from rat hepatocytes (such as the S9 mix used for the Ames test). Respective biotransformation studies will include a fish-specific S9 enzyme assay (OECD 319B) [x2], and the results will feed the PBTK modelling regarding metabolism.”

(13) Line 292, why the only Thyroid related if the paper is specific to TRE then it is better to change the title so that the study could be more specific 

Reply: we are not sure if we understand correctly: does the reviewer mean we should change the title of the paper, because we focus on the TH system? As outlined above, the TH system was selected as a proof-of-principle for development of a cross-class AOP and breaking down the wall between environmental and human health testing. The title of the paper is the title of our project, which has been accepted and approved for EU funding. We do not want, and we cannot change this. Each EURION partner will provide one paper for this special issue and they will all have the names of the respective projects.

Line 367, It is better to discuss more on this section as it does not qualify to understand every aspect used in this part related to the physicochemical domain

Reply: Addressed. The updated version now explains the role of bioavailability for in vitro bioassays, and how this is affected by the selected physicochemical properties:

“An important issue regarding in vitro bioassays are factors controlling the bioavailability of test substances. In this context, compound dissolution may be hampered by sorption (to vials as well as to extracellular matrix) and volatilization, which in turn are driven by hydrophobicity and Henry’s law constant, respectively. Moreover, these physicochemical properties affect both toxicokinetics and toxicodynamics, considering the variation in compound affinity for water-rich vs water-poor tissues (compartments), pathways and receptor sites. From this viewpoint, it is of interest to profile the physicochemical space covered by our ERGO reference compounds.”

“..This wide physicochemical profile ..”

We simplified Fig 6 by removing the MIE´s because we were not confident with the quality of the validation of the presented MIE´s and also not confident that all potential MIE´s were included. This has also been reflected in Appendix A where the MIE´s are removed.

(14) Expected impact, I think it is better to write insilico first then goes on invitro and invivo

Reply: we think it makes more sense this way because in silico models are based on in vivo and in vitro information’s.

(15) Please check some of the references have not been according to the journal guideline, follow the journal guideline. 18,31, 33 and many more

Reply: has been changed accordingly with abbreviated journal names and names of first two authors

Reviewer 2 Report

This Project Report focus on mammalian and non-mammalian vertebrate regulatory testing of EDs by identifying, developing and aligning thyroid-related biomarkers and endpoints. Altough it is well organized and informative, it would be better framed in an environmental journal (eg: IJERPH or other), more than in a journal at the molecular level.

Other comments:

L.4, 34 – disruptors

L.35-39 – Aims of study: please use past tense for verbs

L.58 – filling instead of closed

L.61-63 – Please rephrase sentences for a better readability.

L.109 - This topic doens´t necessary improve animal welfare.

L.111- Assessment at the vitro scale provides new clues for automation…

Author Response

Reviewer #2

This Project Report focus on mammalian and non-mammalian vertebrate regulatory testing of EDs by identifying, developing and aligning thyroid-related biomarkers and endpoints. Altough it is well organized and informative, it would be better framed in an environmental journal (eg: IJERPH or other), more than in a journal at the molecular level.

Reply: Thank you very much for the appreciation of our work. The paper was submitted upon invitation to the special issue “Advances in Research of Endocrine Disrupting Chemicals”. All EURION projects will be presented in this special issue.

Other comments:

L.4, 34 – disruptors

Reply: has been changed accordingly

L.35-39 – Aims of study: please use past tense for verbs

Reply: thanks for pointing this out. We have carefully checked the use of past, present and future tense through-out the manuscript, since we are describing a 5-year project, which has been running for 1 year by now. Thus, for some parts, the work has not yet started. Especially the development of the AOP network described here, is not accomplished yet and therefore, we prefer to use future tense.

L.58 – filling instead of closed

Reply: has been changed accordingly

L.61-63 – Please rephrase sentences for a better readability.

Reply: thanks for pointing that out. We have separated and shortened the sentences for better readability.

L.109 - This topic doens´t necessary improve animal welfare.

Reply: thanks for addressing this inadequate statement. We changed the sentence to “thus complying with the 3Rs principle”, which is what we actually wanted to express.

L.111- Assessment at the vitro scale provides new clues for automation…

Reply: has been changed accordingly

Round 2

Reviewer 1 Report

All comments have been revised in the paper.

Reviewer 2 Report

Authors addressed all the questions and the MS was improved.